Indirect effects of overfishing on Caribbean reefs: sponges overgrow reef-building corals

Loh Tse-Lynn 1 *
McMurray Steven E. 1
Henkel Timothy P. 2
Vicente Jan 3
Pawlik Joseph R. 1 pawlikj@uncw.edu
1 Department of Biology and Marine Biology and Center for Marine Science, University of North Carolina Wilmington , Wilmington, NC , USA
2 Department of Biology, Valdosta State University , Valdosta, GA , USA
3 Institute of Marine and Environmental Technology, University of Maryland Center for Environmental Science , Baltimore, MD , USA
Gandini Patricia
* Current affiliation: Daniel P. Haerther Center for Conservation and Research, John G. Shedd Aquarium, Chicago, IL, USA

Electronic publication date: 2015 Apr 28
Publication date: 2015
Volume: 3
Electronic Location ID: e901
Received 2015 Feb 13; Accepted 2015 Mar 31
Copyright: © 2015 Loh et al.
Copyright year: 2015
Copyright holder: Loh et al.
License: This is an open access article distributed under the terms of the Creative Commons Attribution License, which permits unrestricted use, distribution, reproduction and adaptation in any medium and for any purpose provided that it is properly attributed. For attribution, the original author(s), title, publication source (PeerJ) and either DOI or URL of the article must be cited.
License URL: https://creativecommons.org/licenses/by/4.0/

Keywords: Food webs, Trophic cascades, Indirect effects, Resource trade-offs, Chemical defense, Top-down control, Spatial competition, Coral reefs, MPAs, Marine protected areas

Funding: AMNH Lerner Gray Fund for Marine Research, UNCW Brauer Fellowship and Graduate Student Association Travel Award, NOAA-NURC NA96RU-0260 NOAA’s Coral Reef Conservation Program and the National Science Foundation OCE-0550468 1029515 Funding for this study was provided by the AMNH Lerner Gray Fund for Marine Research, UNCW Brauer Fellowship and Graduate Student Association Travel Award, NOAA-NURC (NA96RU-0260), NOAA’s Coral Reef Conservation Program and the National Science Foundation (OCE-0550468, 1029515). The funders had no role in study design, data collection and analysis, decision to publish, or preparation of the manuscript.

==============================
Consumer-mediated indirect effects at the community level are difficult to demonstrate empirically. Here, we show an explicit indirect effect of overfishing on competition between sponges and reef-building corals from surveys of 69 sites across the Caribbean. Leveraging the large-scale, long-term removal of sponge predators, we selected overfished sites where intensive methods, primarily fish-trapping, have been employed for decades or more, and compared them to sites in remote or marine protected areas (MPAs) with variable levels of enforcement. Sponge-eating fishes (angelfishes and parrotfishes) were counted at each site, and the benthos surveyed, with coral colonies scored for interaction with sponges. Overfished sites had >3 fold more overgrowth of corals by sponges, and mean coral contact with sponges was 25.6%, compared with 12.0% at less-fished sites. Greater contact with corals by sponges at overfished sites was mostly by sponge species palatable to sponge predators. Palatable species have faster rates of growth or reproduction than defended sponge species, which instead make metabolically expensive chemical defenses. These results validate the top-down conceptual model of sponge community ecology for Caribbean reefs, as well as provide an unambiguous justification for MPAs to protect threatened reef-building corals.

An unanticipated outcome of the benthic survey component of this study was that overfished sites had lower mean macroalgal cover (23.1% vs. 38.1% for less-fished sites), a result that is contrary to prevailing assumptions about seaweed control by herbivorous fishes. Because we did not quantify herbivores for this study, we interpret this result with caution, but suggest that additional large-scale studies comparing intensively overfished and MPA sites are warranted to examine the relative impacts of herbivorous fishes and urchins on Caribbean reefs.

Introduction

Food web dynamics are considered fundamental to the study of ecology (Fretwell, 1987), and are the subject of considerable research despite the theoretical limitations brought by the complexity of natural ecosystems. Policy decisions relevant to the management of living natural resources require an in-depth understanding of ecosystem structure and properties (Hooper et al., 2005; Farber et al., 2006). Among the mechanisms important to ecosystem function are indirect effects, which alter community structure through predation (e.g., trophic cascade) or competition (e.g., indirect mutualism; Wootton, 1994). Indirect effects can be difficult to identify or quantify, particularly for complex ecosystems with demonstrable bottom-up control (Strong, 1992). While a number of examples of indirect effects have been found among both terrestrial and aquatic ecosystems, with the rocky intertidal presenting a particularly well-studied example (Menge, 1995), most of these have been described at the species level rather than at the community level (Polis et al., 2000).

Caribbean coral reefs are strikingly different from those of the Indo-Pacific in having two- to ten-fold greater biomass of sponges (Wilkinson & Cheshire, 1990). Sponges have been ignored in broader discussions of coral reef community ecology, in part because they were considered to be free of top-down control (Randall & Hartman, 1968). However, a survey of sponge chemical defenses against fish predators revealed that both palatable and defended sponge species were found on reefs (Pawlik et al., 1995). Manipulative field experiments demonstrated that palatable species had faster rates of wound healing, tissue growth, and recruitment that act in opposition to grazing by sponge-eating fishes (primarily angelfishes and parrotfishes), while defended species produced defensive secondary metabolites (e.g., Walters & Pawlik, 2005; Pawlik et al., 2008; Leong & Pawlik, 2010). In light of these resource trade-offs, a conceptual model of sponge ecology was proposed that included three trophic levels and indirect effects of sponge competition with reef-building corals (Pawlik, 2011). The consumptive indirect effects of this conceptual model were tested by surveying sites on opposite ends of a spectrum of human fishing intensity on Caribbean reefs (Loh & Pawlik, 2014), where a fortuitous long-term manipulative experiment has been ongoing for decades or longer, with some reefs heavily overfished through the use of non-selective fish-traps and nets (e.g., Jamaica, Martinique, Panama), while others have been relatively protected from fishing, either because of low human population density or through the imposition of marine protected areas (MPAs: Bonaire, Cayman Islands, Southeastern Bahamas Islands). This test of theory was noteworthy not only for its spatial scale, but also because it examined community level differences in chemical defenses of a taxonomically diverse group across a large geographic region, with identification of the palatability of 109 sponge species. Results of the Caribbean-wide survey showed that, at less-fished reef sites with many sponge predators, there was a high abundance of chemically defended sponge species, while overfished sites were dominated by palatable species that have faster rates of growth, reproduction or recruitment (Loh & Pawlik, 2014).

In the present study, we used benthic surveys that were performed contemporaneously with the fish and sponge surveys of the previous study (Loh & Pawlik, 2014) to test the indirect effects of overfishing on competition between sponges and reef-building corals. We predicted that removing the top-down control of sponges by overfishing sponge predators would increase competitive sponge-coral interactions, because faster-growing palatable sponges would dominate in the absence of sponge predators. Our surveys also recorded the abundance of other benthic organisms, including macroalgae, at 69 sites across the Caribbean, providing a snapshot of reef community structure and allowing for comparisons of the relative abundances of competitive benthic groups for sites at the extremes of fishing intensity.

Materials and Methods

To maximize the manipulative effect of fishing pressure, we chose survey sites at the extremes of a gradient of fishing intensity, focusing on overfished sites where fish-traps and nets have been used for decades, and less-fished sites that were either far from anthropogenic impacts, or had been maintained as MPAs. Descriptions and a map of sites have been previously published (Loh & Pawlik, 2014). Surveys of coral reefs were carried out at 69 sites from 12 countries in the Tropical Northwestern Atlantic marine province (“Caribbean”) at depths of 10–20 m, except for six sites in Panama and two sites off Florida, USA that were surveyed at 2–7 m (Loh & Pawlik, 2014). Countries surveyed were the Bahamas Islands, Panama, Bonaire, Curaçao, USA (Florida Keys and Puerto Rico), Martinique, St. Eustatius, St. Lucia, the Dominican Republic, Jamaica, Cayman Islands and Mexico (Yucatan coast). Reef site selection was based on previously published assessments of fishing pressure (Burke & Maidens, 2004), prior to our own surveys of the abundance of sponge-eating fishes and the presence of fish-traps.

At each survey site, spongivorous fishes (all angelfishes and the three dominant parrotfish species in the genus Sparisoma) were counted using the Reef Check Survey Methodology (http://www.reefcheck.org) in a volume 2.5 m on each side of, and 5 m above, four end-to-end 20 m transect lines placed along the same depth profile (total volume above the reef = 2,000 m3). The Spongivore Index (SI) was calculated to correct for differences in the grazing activity of small fishes at overfished sites. For sites where (1) fish were observed to be very small (<10 cm), and (2) nets or fish-traps were observed, the SI was calculated by dividing the total fish abundance by 10; for all other sites, the SI was equal to the total number of fish counted (Loh & Pawlik, 2014). We consider the SI to be a highly conservative measure of fish grazing activities, because literature-based grazing estimates have compared the impact of one large parrotfish (>25 cm TL) to 24 small (5–10 cm TL) parrotfishes (Fox & Bellwood, 2007), and one large (35 cm) to 75 small (15 cm) parrotfishes (Lokrantz et al., 2008).

At the same sites where fish abundance was counted, benthic community surveys were carried out by evenly placing a 1 × 1 m quadrat 5 times along each 20 m transect line, with 5 replicate transect lines laid end-to-end at similar depth, and a gap of 5 m between each transect (total of 25 quadrats per survey site). The benthos under 25 points within each quadrat were classified into the following categories: reef-building coral, sponge, fire coral (Millepora sp. C Linnaeus, 1758), gorgonian, zoanthid, other benthos, bare rock or dead coral, rubble, sand, silt, macroalgae (all erect species, but primarily Dictyota JV Lamouroux, 1809; Halimeda JV Lamouroux, 1812; Lobophora J Agardh, 1894; and Microdictyon spp. Decaisne, 1841), turfs (including cyanobacterial mats), and coralline algae. A total of 625 points were recorded at each survey site (Table S1). Coral-sponge interactions were quantified within the same number of quadrats along the same transect lines. For all coral colonies with at least 50% of their surface areas within each quadrat, we counted coral colonies in 3 categories: (1) those having no contact with sponges, (2) those that were growing adjacent to and in contact with sponges, and (3) those that were overgrown by sponges such that sponge tissue was covering live coral tissue.

The percentage of coral colonies having no contact with sponges, growing adjacent to sponges, and overgrown by sponges at each site were plotted in a non-metric multi-dimensional scaling (nMDS) ordination with Bray-Curtis distances, followed by ANOSIM (analysis of similarity) to compare coral-sponge interactions (Clarke, 1993). Benthic occurrence data (number per 625 points per transect site) were square-root transformed for an nMDS ordination, and individual variables were then correlated with the scores of axes 1 and 2. ANOSIM was used to compare benthic occurrences between overfished and less-fished sites, with SIMPER (percentage similarity) to determine which benthic categories contributed most to group differences (Clarke, 1993). Additionally, we performed linear regressions to examine the effect of SI on cover of palatable sponges, the percentage of coral overgrown by sponges, and macroalgal cover, and to relate cover of palatable sponges with coral overgrowth. All analyses were carried out in R v2.15.2 and PRIMER v6.

Results

The mean Spongivore Index (SI) for less-fished sites was 42.5 ± 2.8 (SE) within the survey volume of 2,000 m3 (n = 44 sites), while overfished sites had a mean SI of 2.1 ± 0.3 per 2,000 m3 (n = 25 sites). Coral colonies on reefs that were less impacted by fishing (n = 22,827 colonies, 44 sites) had less interaction with sponges, with 12.0% of colonies growing either adjacent to sponges (8.8 ± 0.9%) or overgrown by sponges (3.2 ± 0.5%). The incidence of coral-sponge interactions was more than double on overfished reefs (n = 11,278 colonies, 25 sites), with 25.6% of corals growing next to sponges (14.9 ± 1.5%) or overgrown by sponges (10.7 ± 2.9%) (Figs. 1 and 2). Accordingly, in a non-metric multi-dimensional scaling (nMDS) plot of sponge-coral interactions, survey sites assembled into two groups (stress = 0.02, Fig. 3): (1) sites with higher proportions of coral-sponge interactions and lower spongivore abundance (e.g., Jamaica, Martinique, Panama); and (2) sites with corals that were less frequently in contact with sponges and higher spongivore abundance (e.g., Bonaire, Cayman Islands, Florida Keys). Analysis of similarity (ANOSIM) between overfished (n = 25) and less-fished (n = 44) reefs indicated that coral-sponge interactions and the density of sponge-eating fishes were significantly different at p = 0.002, with a Global R of 0.17.

Figure 1 Overgrowth of corals by sponges.

Brain coral Diploria labyrinthiformis C Linnaeus, 1758 overgrown by the most abundant Caribbean sponges in the chemically defended category (A) Aplysina cauliformis, and in the palatable category (B) Mycale laevis. (Hogsty Reef, Bahamas; Bocas del Toro, Panama, respectively). Photographs by Joseph Pawlik (1A) and Tse-Lynn Loh (1B).

Figure 2 Coral-sponge interactions for reef sites that were less-fished (n = 44) and overfished (n = 25).

Mean percentage of coral colonies surveyed that were growing adjacent to, or overgrown by, sponges. Error bars denote standard errors.

Figure 3 nMDS plot of survey sites relating the percentage of coral colonies that had no interaction with sponges, growing adjacent to sponges and overgrown by sponges at each site.

Sites labeled black are less-fished and sites labeled red are overfished. Factors labeled in blue (Overgrown, Adjacent, No interaction). Letter of site code denotes the following locations: B, Bahamas; C, Cayman Islands; D, Dominican Republic; E, St. Eustatius; F, Key Largo, FL; J, Jamaica; M, Martinique; O, Bonaire; P, Bocas del Toro, Panama; R, Puerto Rico; S, St. Lucia; U, Curaçao; X, Mexico.

On less-fished reefs with high abundances of sponge-eating fishes, most of the sponges that overgrew corals were slow-growing, chemically defended species (70.9%), reflecting their greater abundance on reefs where predation pressure is high (Loh & Pawlik, 2014). The chemically defended Aplysina cauliformis HJ Carter, 1882 (Fig. 1A), also the most common sponge on Caribbean reefs (Loh & Pawlik, 2014), had the highest number of encounters with corals, accounting for 14.3% of overgrowth interactions (Table 1). On overfished reefs, 43.2% of the sponges that overgrew corals were the faster-growing, palatable species (Loh & Pawlik, 2014), with the palatable sponge Niphates erecta P Duchassaing & G Michelotti, 1864 most frequently recorded overgrowing corals (9.7%, Table 1). Sponges with unknown chemical defense strategies accounted for only 0.2% and 0.1% of sponges overgrowing corals on less-fished and overfished reefs, respectively.

Table 1 Percentage of the ten most common sponge species overgrowing reef-building corals on less-fished and overfished reefs, indicating the chemical defense category of each species.

Less-fished	Overfished	
Species	%	Defense	Species	%	Defense	
Aplysina cauliformis	14.29	D	Niphates erecta	9.72	P	
Mycale laevis	12.44	P	Amphimedon compressa	8.34	D	
Ircinia felix	6.76	D	Aplysina cauliformis	8.17	D	
Svenzea zeai	6.45	D	Mycale laevis	8.08	P	
Amphimedon compressa	5.07	D	Chondrilla nucula	7.66	P	
Agelas citrina	3.84	D	Iotrochota birotulata	5.42	P	
Xestospongia muta	3.38	P	Xestospongia proxima	4.91	P	
Aplysina fistularis	3.07	D	Aplysina fulva	4.82	D	
Aiolochroia crassa	2.76	D	Amphimedon erina	2.75	D	
Niphates erecta	2.76	P	Haliclona walentinae	2.58	D	
Notes.

D, chemically defended, or P, palatable (including chemically undefended and variably defended species). Defense category based on previous research (Pawlik et al., 1995; Loh & Pawlik, 2014).

Linear regression analysis of all sites confirmed that palatable sponge cover was negatively correlated with SI (p < 0.001; r2 = 0.280; Fig. 4A). Also, linear regression analysis indicated that a higher percentage of coral colonies were overgrown by sponges as the cover of palatable sponges increased (p < 0.001, r2 = 0.551). Correspondingly, there was a significant negative relationship between the percentage of corals overgrown by sponges and SI (p = 0.010, r2 = 0.095, Fig. 4B).

Figure 4 Linear regression plots of benthic cover or overgrowth vs. SI.

(A) Palatable sponge cover, (B) percentage of corals overgrown by sponges and (C) macroalgal cover vs. SI. Cover is defined as the number of occurrences in 625 benthic survey points at each site.

From our benthic surveys, macroalgae comprised the most abundant benthic organisms on Caribbean coral reefs, with an overall cover of 28.6%. Sponges and reef-building corals were next with total cover of 15.9% and 16.2%, respectively (composition of benthos by survey site listed in Table S1). Reef-building corals were more abundant on reefs off Bonaire, Curaçao, the Dominican Republic, and Panama, with cover ranging from 22.1–33.3% by location. At other locations, coral cover was less than 15%. The highest cover of macroalgae by location was found on overfished reefs off Jamaica (15.4–68.0%, mean = 50.4%). However, sites having abundant sponge-eating fishes, such as Mira Por Vos Cays (Bahamas, 50.6%), Lac Cai (Bonaire, 36.2%), Banco Chinchorro (Mexico, 14.2–54.9%, mean = 39.3%), the Cayman Islands (35.2–51.7%, mean = 45.1%) and Desecheo Island (Puerto Rico, 50.2%), also had high macroalgal cover.

While all less-fished sites grouped together in the nMDS, several overfished sites had benthic communities similar to less-fished sites (stress = 0.19, Fig. S1). Sponge and zoanthid cover was inversely correlated with Axis 1 (r = − 0.86 and −0.74 respectively), while macroalgal cover was positively correlated with Axis 1 (r = 0.80) (Table S2). For Axis 2, sites were sorted based on turf (r = 0.86) and rock cover (r = − 0.64). Based on correlations with the ordination axes, reef-building coral cover did not contribute to the overall variation in community composition among survey sites (r = − 0.16 and 0.08 respectively). From the ANOSIM, the benthic communities at less-fished sites were significantly different from overfished sites at p = 0.001, with a Global R of 0.34. Percentage similarity (SIMPER) analysis showed that less-fished sites were characterized by higher macroalgal, rock, reef-building coral and coralline algal cover, and less turf and sponge cover (Table 2). Linear regression analysis of all sites also indicated that SI was not correlated with macroalgal cover (p = 0.528, r2 = 0.006; Fig. 4C).

Discussion

Sponge overgrowth of corals was greater on overfished reefs

From the standpoint of Caribbean coral reef conservation, our study provides compelling justification for fishing restrictions to protect sponge-eating fishes (angelfishes and parrotfishes) in order to decrease competitive interactions between reef-building corals and sponges. The three-fold difference in overgrowth of corals by sponges between less-fished and overfished sites was substantial, particularly when over 25% of coral colonies at overfished sites were in contact with, or overgrown by, sponges. In a previous study, we demonstrated that a palatable sponge species, Mycale laevis HJ Carter, 1882, was restricted to refuge habitats when sponge-eating fishes were abundant, but overgrew living coral tissue when sponge predators were absent or rare (Loh & Pawlik, 2012) (Fig. 1B). Here, we were able to observe this phenomenon at the community level and across an entire geographic region. The competitive superiority of sponges over reef-building corals has been well documented, and is likely due to a combination of shading, physical inhibition of water flow and gas exchange (smothering), and the use of allelopathic secondary metabolites to kill coral tissue (Porter & Targett, 1988; Thacker et al., 1998; Aronson et al., 2002; Pawlik et al., 2007) (Fig. 1). Because allelopathic metabolites are present in the mucus or exudates of some sponge species, mere proximity to reef-building corals may be sufficient to negatively impact coral physiology and reproduction, making affected colonies more susceptible to bleaching or pathogenesis (Sullivan, Faulkner & Webb, 1983). With the recent announcement that five species of Caribbean reef-building corals are proposed for listing as “threatened species” under the United States Endangered Species Act (NOAA , 2014), the results of this study should be useful in justifying regulations to protect sponge-eating fishes.

This study underscores the distinctive ecology of Caribbean coral reefs relative to those in other parts of the world, a concept that is not new (Wilkinson & Cheshire, 1990; Roff & Mumby, 2012), yet often unacknowledged in reviews of coral reef ecosystem function. Sponges dominate benthic communities on Caribbean coral reefs to a greater degree than elsewhere, but this fact is usually obscured by sampling methods. Coral reef ecologists conventionally survey 2-dimensional benthic cover because of the time constraints of scuba diving and the complexity of reef topography. While overall cover of sponges from our surveys was nearly the same as corals (15.9 vs. 16.2%), and well behind macroalgae (28.6%), both reef-building corals and macroalgae consist primarily of thin layers of tissue intended to catch light for photosynthesis. The filter-feeding sponges recorded in these surveys were mostly thick-bodied, and in many cases massive or upright branching species, so that the actual biomass of sponges on Caribbean reefs (from reef crest to deep mesophotic reefs and including reef interstices) is likely to be orders of magnitude greater than that of algae or corals. Sponge communities are structured by top-down processes, but may be a rare example of a system unaffected by bottom-up factors (Pawlik et al., 2013; Pawlik et al., 2015). The primary reason for this may be the nutritional reliance of Caribbean reef sponges on dissolved organic carbon (DOC), which frees sponges from food-limitation and provides a trophic “loop” that returns refractory DOC from the water column to the benthos (De Goeij et al., 2013). A similar nutritional strategy does not appear to be available to sponges on more oligotrophic Indo-Pacific coral reefs (Wilkinson & Cheshire, 1990).

Macroalgal cover on overfished and less-fished sites

An ancillary outcome of the benthic surveys conducted for this study was the surprising result that macroalgal cover was not lower on less-fished reefs. Linear regression revealed no relationship between SI and macroalgal abundance (Fig. 4C), and the SIMPER analysis indicated that overfished reefs had lower macroalgal cover (Table 2). When reef sites were split based on SI (as in Fig. 2) mean percentage cover of macroalgae was significantly higher on less-fished than overfished reefs (38.1 vs. 23.1%; one-tailed t-test on arc-sine transformed data, p = 0.044). It is generally understood that a greater abundance of herbivorous fishes correlates with less macroalgal cover (e.g., Knowlton & Jackson, 2008), and one wide-ranging survey of Caribbean reefs has supported this view (e.g., Newman et al., 2006). Considering the methods used in this study, how valid is this contrary outcome?

Table 2 SIMPER dissimilarity matrix for square-root transformed occurrences of benthic categories between less-fished and overfished sites.

Average dissimilarity = 32.77	
	Less-fished	Overfished					
Benthic category	Average abundance	Average abundance	Average dissimilarity	Dissimilarity SD	% Contribution	Cumulative %	
Macroalgae	13.56	10.02	5.34	1.41	16.29	16.29	
Turf	6.85	7.02	3.94	1.44	12.03	28.32	
Sponge	8.17	11.45	3.43	1.17	10.47	38.79	
Rock	8.00	5.78	3.32	1.47	10.12	48.91	
Hard coral	9.84	8.99	3.03	1.42	9.24	58.15	
Coralline algae	6.49	3.41	2.58	1.42	7.88	66.04	
Gorgonian	3.58	2.69	2.24	1.35	6.84	72.88	
Sand	5.47	5.51	2.04	1.37	6.21	79.09	
Silt	0.86	2.61	1.79	0.89	5.47	84.56	
Rubble	2.56	3.22	1.70	1.18	5.18	89.74	
Fire coral	1.41	1.19	1.15	1.00	3.50	93.25	
Notes.

% contribution indicates the contribution to dissimilarity between less-fished and overfished groups.

Unlike sponges, macroalgae may undergo seasonal changes, with low abundance in the winter (Lirman & Biber, 2000). Of the 69 surveys performed for the present study (Dataset S1, Loh & Pawlik, 2014), most were carried out during the summer and early fall (June–October) when macroalgal cover is high. Only 3 surveys were performed in the winter (Florida Keys sites F1–F3), but these had relatively high macroalgal cover for 2 of 3 sites (33, 6, 22%). Hence, there was no evident bias in the seasonal timing of surveys that would explain the observed relationship between fish abundance and macroalgal cover.

Spongivorous fishes were surveyed for the present study, not herbivores. It could be argued that, in the absence of a full accounting of herbivorous fish species, any relationship between fish abundance and macroalgal cover is ambiguous. However, the 25 overfished sites surveyed in this study were mostly stripped of fishes larger than the mesh-size of nets and fish-traps, including other herbivorous fishes (primarily Scarus species and acanthurids). It could also be argued that in the absence of size data (and hence, biomass), any relationship between fish abundance and macroalgal cover is equivocal. But, again, we know that the fishes at the overfished sites were both very small and relatively few compared to less-fished sites, due to the fishing methods employed at overfished sites. While it is true that one other wide-ranging survey study documented a negative correlation between fish and macroalgal biomass (Newman et al., 2006), no previous study has targeted intensively overfished sites over as wide a region as reported herein (Jamaica, Panama, Martinique, St. Lucia, Puerto Rico and the Dominican Republic in the present study; only Jamaica in Newman et al., 2006). Rather than a gradient in fishing pressure, as in Newman et al. (2006), the present study targeted the presence and absence of intensive fishing by specifically surveying sites that were intensively overfished and relatively protected from fishing.

The present study is not alone in its conclusions, as other survey studies have noted the absence of a correlation between macroalgal cover and herbivorous fish counts (Lirman & Biber, 2000) and MPA status of reef sites (Toth et al., 2014). Furthermore, higher levels of nutrients from the excretion of reef fishes (as total fish biomass) has been shown to correlate with greater macroalgal cover (Burkepile et al., 2013), a conclusion that is supported by the present study. Comparisons of Caribbean reefs with those of the Indo-Pacific have led some to question the top-down control of macroalgae by herbivorous fishes on the former (Roff & Mumby, 2012). Caribbean reefs suffered the catastrophic loss of the sea urchin Diadema antillarum RA Philippi, 1845 in the early 1980s, and this species may have played a disproportionate role in herbivory (Shulman & Robertson , 1996) relative to what occurs on Indo-Pacific reefs. In the present study, the abundant macroalgal cover at geographically isolated, less-fished sites in the SE Bahamas or Banco Chinchorro, Mexico, could be attributed to higher nutrient addition from total fish biomass, to the continued absence of D. antillarum, or to differences in macroalgal species and palatability among sites. For example, the unpalatable Microdictyon spp. (Lapointe et al., 2004) and Dictyota spp. (Hay, 1991) were common in our surveys of these sites and are generally avoided by fish grazers. While we did not enumerate D. antillarum in this study, it may be that populations of this important herbivore are rebounding faster on overfished reefs where urchin predators have been removed by fish-trapping, along with herbivorous and spongivorous fishes. If true, this may explain the generally lower levels of macroalgae on overfished reefs observed in this study. Despite the limitations of the survey data as discussed above in addressing the relationship between herbivorous fishes and macroalgae on Caribbean reefs, the surprising outcome, combined with the scale of this study, the choice of intensively overfished sites, and very recent reassessments of the impacts of fish herbivores on Caribbean reefs (Adam et al., 2015) argue for its consideration in future, more targeted, survey studies of the impacts of herbivores on reefs.

Conclusions

Validating our conceptual ecosystem model (Pawlik, 2011), Caribbean reef sponges provide a rare example of indirect effects at the community level, in which a group of consumer species (primarily angelfishes and parrotfishes) act upon a diverse community of sponges to alter their relative abundance and thereby change the competitive interactions of the sponge community with reef-building corals. In the present study, indirect effects were propagated from human fishing activities, but this role may have been played by higher-level predatory fishes in the past, likely from two trophic levels (requiem sharks—large groupers and snappers), although probably not as effectively as human fish-trapping removes sponge predators. On the other end of the model, palatable sponges compete with corals on overfished reefs, but also appear to compete with macroalgae, as the abundance of the two were inversely correlated. In contrast to this model system, most commonly cited examples of indirect effects are simple ecosystems with trophic levels often identified as individual species (e.g., orca—sea otter—urchin—kelp; wolf—elk— aspen—songbirds Wootton, 1994; Hebblewhite et al., 2005). Despite the high species-diversity at each level, the clarity of indirect effects observed for the Caribbean reef sponge ecosystem is likely due to the simplicity of the interactions relative to other, particularly terrestrial, ecosystems (Polis et al., 2000): abiotic influences on the system are minimal, top-down effects are dominant, sponge community composition is similar across the entire biogeographic region, insect-equivalent mesograzers are unimportant, and the influences of extinctions and invasions are minimal (Pawlik, 2011; Loh & Pawlik, 2014). The clarity and predictive capability of this model system runs contrary to the perception that recent contributions to the ecological literature have been increasingly complex and decreasing in explanatory power (Low-Décarie, 2014).

Supplemental Information

Figure S1 nMDS plot of survey sites relating benthic community structure using square-root transformed occurrences of benthic categories

Occurrence is the number of times each category appears per 625 points of each transect. Sites labeled black are less-fished, and sites labeled red are overfished. Factors labeled in blue. The following benthic categories were used: ca, coralline algae; cme, fire coral; go, gorgonian; hc, hard coral; ma, macroalgae; ot, other benthos; rc, rock; rb, rubble; s, sand; si, silt; sp, sponges; tu, turf; and zo, zoanthid. Prefixes of site names denote the following locations: B, Bahamas; C, Cayman Islands; D, Dominican Republic; E, St. Eustatius; F, Key Largo, FL; J, Jamaica; M, Martinique; O, Bonaire; P, Bocas del Toro, Panama; R, Puerto Rico; S, St. Lucia; U, Curaçao; X, Mexico.

Click here for additional data file.

Table S1 Benthic cover data for all survey sites. (Excel spreadsheet)

Click here for additional data file.

Table S2 Correlation of individual variables with axes 1 and 2 of the nMDS ordination of benthic category occurrence across all survey sites (Fig. S1)

Click here for additional data file.

We thank collaborators and staff from Aquarius Reef Base, St. Eustatius National Marine Park, Soufrière Marine Management Association (St. Lucia), Smithsonian Tropical Research Institute Bocas station (Panama), CARMABI (Curaçao), Punta Cana Ecological Foundation (Dominican Republic), Discovery Bay Marine Laboratory (Jamaica), Action Adventure Divers (St. Lucia), Scubafun Dive Center (Dominican Republic), Espace Plongée Martinique, Florida Keys National Marine Sanctuary, the governments of the Bahamas, Mexico and the Cayman Islands, and the crew of R/V Walton Smith, who variously facilitated permits and provided valuable logistical and field support. Fieldwork was conducted under Permit FKNMS-2009-126 in the Florida Keys, National Commission on Aquaculture and Fisheries (Comisión Nacional de Acuacultura y Pesca) Permit DAPA/2/06504/110612/1608 in the Yucatan (Mexico), Department of Marine Resources Permit MAF/LIA/22 (Bahamas Islands), and unnumbered permits or research contracts from St. Lucia, the Cayman Islands, and St. Eustatius.

Additional Information and Declarations

Competing Interests

Author Contributions

Field Study Permissions

The authors declare there are no competing interests.

Tse-Lynn Loh conceived and designed the experiments, performed the experiments, analyzed the data, contributed reagents/materials/analysis tools, wrote the paper, prepared figures and/or tables, reviewed drafts of the paper.

Steven E. McMurray, Timothy P. Henkel and Jan Vicente performed the experiments, reviewed drafts of the paper.

Joseph R. Pawlik conceived and designed the experiments, performed the experiments, analyzed the data, contributed reagents/materials/analysis tools, wrote the paper, reviewed drafts of the paper.

The following information was supplied relating to field study approvals (i.e., approving body and any reference numbers):

Fieldwork was conducted under Permit FKNMS-2009-126 in the Florida Keys, National Commission on Aquaculture and Fisheries (Comisión Nacional de Acuacultura y Pesca) Permit DAPA/2/06504/110612/1608 in the Yucatan (Mexico), Department of Marine Resources Permit MAF/LIA/22 (Bahamas Islands), and unnumbered permits or research contracts from St. Lucia, the Cayman Islands, and St. Eustatius.

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
