# Peer review of "Indirect effects of overfishing on Caribbean reefs: sponges overgrow reef-building corals"

_PeerJ, doi:10.7717/peerj.901_

## Round 0.1 · original submission · Major Revisions

· Academic Editor

Major Revisions

It is necessary to follow suggestions made by both reviewers and justify them. The paper is interesting but needs to strengthen some points specifically to address the fact that the herbivory-macroalgal relationship appears to be well beyond your dataset.

·

Basic reporting

The basic reporting is fine, except I suggest the authors include a description of how fish surveys were performed and how the SI was calculated. I understand the methods are published elsewhere but I believe it is preferable to publish them here as well, in part, because the journal they are in (PNAS) is paywalled. Also, could the authors clarify whether the SI is based on density? (I think it is)

"Spongivore Index (SI) was defined as the sum  of total parrotfish and angelfish abundance within a survey volume of 2000 m3 at each site, with total fish abundance divided by 10 at overfished sites to correct for smaller fish biomass (Loh & Pawlik, 2014). "

How were overfished sites identified to make this correction? I thought that the SI level was what determined the degree of fishing.

Experimental design

In general, the experimental design is sound. My only concern is regarding the correction:

"Spongivore Index (SI) was defined as the sum  of total parrotfish and angelfish abundance within a survey volume of 2000 m3 at each site, with total fish abundance divided by 10 at overfished sites to correct for smaller fish biomass (Loh & Pawlik, 2014)"

Was done because only fish density (and not biomass) was measured? Could the authors clarify the rationale for the correction, discuss how it could affect the outcome of the analysis, and justify the use of 1/10 as a correcting multiplier, i.e., why not 0.5 or 1/3?

Validity of the findings

I do think the findings are valid and the inferences made are supported by the empirical results. The increase in sponge-coral encounters with fishing / low SI might seem modest, but I believe it is ecologically relevant. Moreover, coral-algal interactions are also infrequent but can accumulate and affect coral population recovery (see nice work on this by Box and Mumby; they show even when algal cover is low, they impact landscape scale coral recruitment because seaweed location is variable and the eventually zap most of the corals).

The one argument I am skeptical of is this:

"Coral-sponge competition provides an additional and unambiguous justification for marine protected areas (MPAs) in the Caribbean”

Because Caribbean MPAs very rarely effectively conserve or restore fish populations (due mainly to poor enforcement and small size), I don't see them as a realistic solution here.

Additional comments

This manuscript quantifies the relationship(s) between fishes, corals, sponges and algae on Caribbean reefs to test the hypothesis that fishing affects the frequency (and outcome?) of sponge-coral interactions via the loss of fishes that consume and control sponge populations, i.e., does fishing break down an important TMII?

I agree with the general narrative and think that the results support and justify the main inferences that were made. This large-scale survey approach nicely compliments the local-scale experiments the Pawlik lab has been doing. I agree with the authors that fishing-sponge-coral interactions receive very little attention (really just this lab) and that sponges are more abundant at fished and coral-depleted sites (this reality has been lost on some in the sponge world, I think because they view sponges as threatened in the same way corals and some fishes are). Finally, I agree completely (and this is controversial and may upset some reviewers) that evidence for a fishing-algae link is actually pretty weak. Overall, I think the manuscript is expertly written and I appreciate the breadth of the concepts in the Intro and Conclusion.

“On average, overfished sites had lower macroalgal cover, contrary to prevailing assumptions about seaweed control by herbivorous fishes.”

Fascinating.

Reviewer 2 ·

Basic reporting

See General Comments

Experimental design

See General Comments

Validity of the findings

See General Comments

Additional comments

The authors present an interesting companion paper to their PNAS paper from last year where they examine the relationships between the potential for fishing to shift the spongivorous fish community to a place where it facilitates the replacement of defended sponges by palatable sponges and increases competition with corals. I actually find that part of the paper pretty interesting but have a few suggestions below that could improve it.

Where I struggle with the paper is the wide extrapolations as to what their data on algal abundance means in less fished vs overfished areas. They even put in their abstract that their data set challenges the assumption of the impacts of fishing on control of algal abundance on reefs. While they may show more algae in less fished areas (but we don’t know because they didn’t actually test this), they don’t seem to present enough data on the fish community for me to understand how fishing may have impact this pattern. They present data on the spongivore index then relate that to macroalgal abundance but that is pretty marginal. They need better, more straightforward analyses on the relationships between algal and herbivores to be able to say anything concrete. They either need to do this or relegate their speculation on algal control by herbivores to a small point in the discussion. But, given how they counted fishes, they won’t have the data necessary to address tehse questions. As written and analyzed, those points don’t belong in either the abstract or introduction. They just have very little support for the points they are trying to make here.

So, in summary I like how they address the coral-sponge competition angle in relation to spongivores, although this could have been included in the PNAS paper which would have made a really robust paper there. But, they cannot do justice to arguments about macroalgae and herbivory and should avoid that line on inquiry except at the very basic observational level, i.e. macroalgae may be more abundant in less fished areas but can’t really say why.

Specific comments

Abstract – What is the actual difference in macroalgal cover here? Authors never say. Seems disingenuous, bordering on irresponsible, to bring it up here to question herbivore control of seaweed on Caribbean reefs given the cursory analyses done in their data relating algal cover to herbivore abundance.

Introduction – First paragraph is a little vague. But what I am surprised at is the claim that indirect effects have not been well demonstrated at the community level. In fact there are lots of good examples of indirect effects impacting community organization going back to Bob Paine in the 1960’s. I am surprised by this claim. Work from rocky intertidal systems alone have produced dozens of papers on the impact of indirect effects in community organization.

Lines 26-39 - This is a pretty biased argument here. Most of the data from teh Caribbean come from places where herbivory has already been compromised. And Diadema were likely most important in places where herbivorous fishes had been removed, predators had been removed, and Diadema were at ridiculously high densities. I suggest being more evenhanded here in your introduction of herbivory and top down control on Caribbean reefs. Where research has been done in places with lots of herbivores, top down control is strong, Caribbean wide assessments of relationships between herbivores and macroalgae suggest a negative correlation. So i suggest dialing back the hyperbole here about the ineffectiveness of herbivores in the Caribbean. Further, this line of argument isn’t even the basic point of this study. It is at best an ancillary point as they analysis examining the relationship between herbivory and macroalgae are tenuous at best.

Lines 74-81 – I had to go back to the Loh and Pawlik 2009 paper to look up how they were counting their fishes. I have to say I was pretty disappointed with the method. They only have abundance and no size estimates so we don’t know if a parrotfish is 5cm or 25cm which is disturbing. So there can be no good estimate of biomass and how it relates to sponges (or macroalgae) and biomass is a much better predictor than abundance as it is more directly translatable to metabolic rates and consumption. Further, I am unclear why they express their fish counts as fish per 2000m3 a volumetric scale. I almost never see that in the literature. its always on an areal scale (m2, km2, hectares). So expressing their abundance like this makes it very difficult to compare across different studies. My biggest reservation is that they only counted spongivores which means they only counted Sparisoma parrotfishes and not Scarus parrotfishes. So they are missing a whole genus of parrotfish not to mention surgeonfish when they try to make arguments relating fishing to herbivory to algal abundance. Further, they have no estimates of predator abundance and we know that the abundance of larger predators can have both consumptive and non-consumptive effects on herbivores (and likely spongivores). So I was really disappointed at the level of characterization in the fish community.

As a side note, the big Scaus spp parrotfishes are really important spongivores. Our data from following those guys around shows that up to 5-10% of the bites of Scarus coelestinus and Sc. guacamaia come from sponges. So studies moving forward off of this should count fishes better and more completely.

Lines 105-109 – this seems like a really deterministic NMDS given that these numbers from sponge occurrence will almost be the reciprocals of each other. There are only three categories and they will add up to 100%. So no wonder the less fished and overfished reefs break out so cleanly. What might be a more informative analysis is if the NMDS categories included contact by palatable/defended sponges rather than just sponges. Also, having SI in the NMDS seems wrong given that the categories are already less fished vs overfished which is based on the SI. Why isn’t the analysis just with the sponge data here?

Lines 109-110 - Is this occurrence data (presence/absence) or percent cover data? those are different things.

Lines 123-125 - One of the things that strikes me in Fig 1 is that both of the interactions in panels A and B were likely counted as ‘overgrowth’. But the interaction in B is clearly more intense competition on the coral that in A. There is actually minimal competition in A. So what I would have liked to have seen is some index of competition. Were the overgrowth interactions in fished areas even more intense? If sponges like the Aplysina were doing most of the interactions in less fished areas then maybe those interactions are even less harmful because of the ropy growth form as opposed to something like Mycale that is big and fat and grows right next to the coral tissue. If the authors have any data on potential intensity of competition, it should definitely be included.

Fig 4A – Those data just don’t look amenable to linear regression they are so zero inflated (or rather whatever the minimum SI is).

Lines 160-163 - Overlaying the vectors of the original dataset here would be useful for visualizing how the different benthic components were related to teh spatial separation of the sites.

Lines 169-170 - So why not test directly the difference between macroalgal cover in less fishes vs overfished reefs? I don’t see that they do that. Nor do they report what these means are. Yet they make this a big point in their abstract about overfishing not affecting algal abundance which is concerning given the potential implications.

Lines 170-171 - What about just biomass of parrotfishes here? Why would the SI index relate to macroalgal cover at all? Yes it is parrotfish based but rather use the raw data than some contrived index.

Lines 194-214 - I am really uncomfortable with the interpretation of the relationship between macroalgae and fishing pressure here. First, they really only have a cursory assessment of the herbivore community. They only survey parrotfishes, don’t present biomass, and don’t do any analyses on the relationship between parrotfishes and macroalgae as far as I can tell. One test between macroalgae and SI but that isn’t very meaningful.

They cite the Newman paper here, but the Newman paper was a robust characterization of the whole fish community and the benthos. Further the Newman paper did rigorous tests of the relationships between the benthos patterns and fish communities. To position this current dataset as contrary to that Newman one is a little irresponsible given the cursory level of the analyses here.

Lines 201-202 - The should not be confident in this. This should have been measured. When I looked back at how they measured fishes, they didn’t even count all of the parrotfishes, only Sparisoma parrots and not Scarus spp, because Sparisoma are the spongivores. So they are missing a genus of parrotfish, all the surgeonfish, etc in their estimates of herbivores.


Lines 232-233 - This seems to be a gross oversimplifcation. Indo-Pacific reefs vary broadly in complexity, nutrient levels, DOC availability etc just like Caribbean reefs.
Bonaire and Bocas del Toro reefs are both ‘Caribbean’ reefs but certainly very different with different ecological forces structuring there communities, likely their sponge communities as well given the high sedimentation and POM in the water in Bonaire. So these kinds of simplified statements that are rife in the MS are frustrating.

Lines 235-236 - What do they mean ‘rare example of indirect effects at teh community level’. There have been good examples of indirect effects structuring ecological communities for five decades.

Further, the authors don’t actually show the indirect effect on community organization. They have the direct link (correlation) between fishing and increased sponges and changes in the sponge community. But then they link this to increased interactions with corals. There is no link to changing the abundance of coral or declines in coral recruitment, growth, etc. So the indirect effect is not there although they do show the potential is there. So they need to be more speculative with their arguments here.

Lines 244-249 - But they are ignoring good coral reef examples as well.

Hughes et al. 2007 Current Biology - herbivores excluded, macroalgae grow, corals decline

Burkepile and Hay 2008 PNAS - herbiore identity changed, macroalgal community changes, corals decline

Newman et al. 2006 Eco Lettts - a paper they cite - a more detailed analysis of Caribbean benthic communities than they do here and showed the indirect interaction chain that the authors here suggest of links between herbivores, algae, and corals

So bottom line is that this dataset here is not as unique as the authors are trying to make it out to be.

Table 1 - So do these numbers represent the % of the sponge community or the % cover of the benthos of each of these species?

Table 2 - So by my addition, these groups only add up to ~60-70% cover for both reef types (sums of first two columns here)? So they are missing 30% cover here. What else is covering 30% of the benthod? This Table 2 doesn’t seem to jive with the data in their Table S1 either where the average macroalgal cover is ~26% overall and 31% for less fished and 23% for overfished reefs.

Fig 1 - tough to say that the sponge is ‘overgrowing’ the coral in panel A. yes there is some minimal contact but nothing like the clear overgrowth and intense competition in panel B.

Fig 3 - would be interesting to overlay the vectors of the different pieces of data going into this so that we can see how the different data goes into driving separation of the different sites.

---

## Round 0.2 · accepted · Accept

· Academic Editor

Accept

Authors have undertaken a major revision of the manuscript. Methods has been fully revised and discussion has been changed according to Reviewer concerns. This is a very interesting paper, congratulations.